# Building blocks and blueprints for bacterial autolysins

**Spencer J. Mitchell**[1], **Deeptak Verma**[2], **Karl E. Griswold**[3,4], **Chris Bailey-Kellogg**[1,4]*

**1** Department of Computer Science, Dartmouth, Hanover, New Hampshire, United States of America, **2** Computational and Structural Chemistry, Merck & Co., Inc., Kenilworth, New Jersey, United States of America, **3** Thayer School of Engineering, Dartmouth, Hanover, New Hampshire, United States of America, **4** Lyticon LLC, Lebanon, New Hampshire, United States of America

* cbk@cs.dartmouth.edu

**Data Availability Statement:** The scripts used to collect and analyze the data, along with the data itself, are at https://git.dartmouth.edu/cbklab/ledgos.

## Abstract

Bacteria utilize a wide variety of endogenous cell wall hydrolases, or autolysins, to remodel their cell walls during processes including cell division, biofilm formation, and programmed death. We here systematically investigate the composition of these enzymes in order to gain insights into their associated biological processes, potential ways to disrupt them via chemotherapeutics, and strategies by which they might be leveraged as recombinant anti-bacterial biotherapies. To do so, we developed LEDGOs (lytic enzyme domains grouped by organism), a pipeline to create and analyze databases of autolytic enzyme sequences, constituent domain annotations, and architectural patterns of multi-domain enzymes that integrate peptidoglycan binding and degrading functions. We applied LEDGOs to eight pathogenic bacteria, gram negatives *Acinetobacter baumannii*, *Klebsiella pneumoniae*, *Neisseria gonorrhoeae*, and *Pseudomonas aeruginosa*; and gram positives *Clostridioides difficile*, *Enterococcus faecium*, *Staphylococcus aureus*, and *Streptococcus pneumoniae*. Our analysis of the autolytic enzyme repertoires of these pathogens reveals commonalities and differences in their key domain building blocks and architectures, including correlations and preferred orders among domains in multi-domain enzymes, repetitions of homologous binding domains with potentially complementarity recognition modalities, and sequence similarity patterns indicative of potential divergence of functional specificity among related domains. We have further identified a variety of unannotated sequence regions within the lytic enzymes that may themselves contain new domains with important functions.

## Author summary

Bacteria use enzymes called "autolysins" to remodel their cell walls, for example during cell division and formation of biofilms. We have developed a bioinformatics pipeline to analyze the autolysin repertoires of a number of different pathogenic bacteria, comparing and contrasting the domain building blocks from which their autolysins are constructed, the amino acid sequence diversity of the domains, and the architectural patterns by which the domains are assembled into complete enzymes. Our analysis provides insights into important biological processes, indicates sequences that would likely benefit from

**Funding:** This work was supported by US National Institutes of Health (https://www.nih.gov/) grant 1R01AI123372 to KEG and CB-K. The funders had no role in study design, data collection and analysis, decision to publish, or preparation of the manuscript.

**Competing interests:** I have read the journal's policy and the authors of this manuscript have the following competing interests: KEG and CB-K are member-managers of Lyticon LLC. No other authors have a conflict of interest. Potential conflicts of interest for KEG and CB-K are under management at Dartmouth. The authors declare that the work presented here is free of any bias.

experimental characterization, highlights potential targets for chemotherapeutic intervention, and even suggests strategies for building next-generation autolysin biotherapies that turn a pathogen's own proteins against it to destroy its cell wall.

## Introduction

Lysins, enzymes that degrade bacterial cell wall peptidoglycan, appear throughout the tree of life. For example, bacteriophage use endolysins to enter their bacterial hosts and exolysins to escape them, bacteria employ bacteriocins in "bacterial warfare" against each other, and animals utilize lysozymes as part of their innate immune defenses against bacteria. Biotechnology has also enabled the potent cell wall hydrolyzing functions of lysins to be leveraged in a wide range of applications. Naturally, lysins are being used as next-generation antibiotics [1–4], with numerous advantages over small molecule antibiotics including catalytic degradation of bacterial cell wall and resultant high potency and rapid onset of action. Examples of lysin-based therapeutics include human lysozyme [5], the bacteriocin lysostaphin in both wild-type [6] and deimmunized [7] forms, and the phage endolysin-based drugs named CF-301 [8] and SAL200 [9]. Even gram-negative bacteria may be susceptible to lysin-based drugs, with help from delivery systems [10,11]. Beyond therapeutics, applications of lysins include food safety and processing, where, for example, lysins are being studied as additives to eliminate pathogens including *Staphylococcus*, *Listeria*, *and Clostridium* in dairy and meat products [12–15]. Lysins are also more generally being used as disinfectants in food safety and other applications, in particular leveraging their ability to clear dangerous biofilms [16–18].

In addition to trans-acting lysins, there are also diverse cis-acting lysins, or autolysins. Bacteria use autolysins to degrade their own cell walls as part of their natural physiology, in processes including cell division, autolysis (programmed death), peptidoglycan recycling, and biofilm formation [19–27]. Autolysins have not been as extensively studied as other lysins, and likewise have not been as extensively utilized in applications, though proof of principle of autolysin-based drugs [28–31] and autolysin usage in food safety [32] have both been established.

Lysins carry out their cell wall degrading functions by recognizing and digesting bonds holding together cell wall peptidoglycan, a mesh of amino acids and glycans. In some cases, a catalytic domain (CAT) handles both recognition and digestion, while in other cases, a lysin also includes one or more cell wall binding domains (CWB) to specifically recognize peptidoglycan. Different CATs attack different peptidoglycan bonds in different contexts; e.g., MurNAc-LAAs target a bond in the stem peptide, N-acetylmuramidases cleave the glycan backbone, and peptidases digest amino acid bonds [21]. Likewise, different CWBs have different recognition specificities; e.g., CW_binding_1 shows affinity for cell walls containing choline [33], while LysM is quite diverse but generally recognizes N-acetylglucosamine [34]. Finally, there is significant diversity in lysin architectures, i.e., how CATs and CWBs are integrated in a single enzyme to work together in attacking peptidoglycan. For example, phage endolysins LysSA12 and LysSA97 consist of a cysteine, histidine-dependent amidohydrolases/ peptidase (CHAP), amidase, and CWB, where the CHAP domain confers the main catalytic activity while the amidase domain confers improved peptidoglycan binding in combination with the CWB [35].

Inspired by the extensive natural and artificial utility of lysins, along with the diversity of CAT and CWB functions, specificities, and architectural patterns, we have set about systematically finding and characterizing autolysin building blocks and architectures within and across a set of important pathogenic bacteria. We are focusing on autolysins, since trans-acting lysins

have been more thoroughly studied and catalogued [36–38], and since autolysins can both provide insights into natural bacterial processes as well as support significant applications. The value of viewing CATs and CWBs as modular "building blocks" has been established in the context of engineering new biotherapeutics [28,39]. More generally, novel autolysins present numerous advantages both as antibiotic targets as well as antibiotics, because these enzymes are integral to bacterial physiology and evolving resistance could disrupt natural biological processes. For example, identification of key autolysin domains might suggest new chemotherapeutic drug targets; e.g., disruption of autolysins LytN and Atl have been shown to result in structural damage to the cell wall, altered cellular morphology, and marked growth defects [22,26,40–42]. Likewise, using autolysins as antibiotics would require the bacterium to evolve entire cellular processes so as to simultaneously modify the cell wall to evade the antibiotic enzyme's function but maintain the endogenous autolysins' functions. Furthermore, changing the cell wall structure may prove detrimental in other ways, as has been shown for the trans-acting lytic enzyme lysostaphin whose resistance phenotype sensitizes *S. aureus* to traditional lactam antibiotics [43–45].

By comprehensively cataloging autolysin repertoires, we seek to characterize patterns that are "interesting" and "important". For example, a domain that is highly conserved in a bacterium may be lethal if knocked out or, as discussed above, may be hard to evolve resistance to if used exogenously as a biotherapeutic. The relative uniqueness of a domain or architecture to a particular species may indicate whether associated drugs would be broad or narrow spectrum. Identification of repeated domains and their relative sequence conservation may point to potentially divergent specificities, or perhaps to the use of avidity in peptidoglycan recognition. Understanding the "blueprints" of how domains are connected in natural lytic enzymes may support construction of potent synthetic enzymes by analogy or generalization, may allow for engineering of multi-functional therapeutic modalities, and may enable inference of functional roles of domains or sequence regions that are not yet well characterized.

With these motivations in mind, we here summarize our pipeline for finding and analyzing autolysin building blocks and architectures, and we illustrate, in application to eight different pathogenic bacteria, how this approach offers multi-faceted insights into these important enzymes.

## Results

The LEDGOs (lytic enzyme domains grouped by organism) data collection pipeline, database, and analysis tool (Fig 1) enables multi-faceted characterization of the diversity of lytic enzyme domains and architectures in bacterial proteomes. We used LEDGOs to collect and analyze common CAT and CWB domains and associated architectures in eight important pathogenic bacteria: four gram negative (*Acinetobacter baumannii*, *Klebsiella pneumoniae*, *Neisseria gonorrhoeae*, and *Pseudomonas aeruginosa*) and four gram positive (*Clostridioides difficile*, *Enterococcus faecium*, *Staphylococcus aureus*, and *Streptococcus pneumoniae*). These bacteria have been identified as urgent or serious antibiotic resistant threats by the CDC [46], making them important candidates for new antibiotics, including those that target their autolysins and those that use their autolysins against them. Table 1 summarizes the composition of the resulting LEDGOs database. The following sections then summarize analyses and insights enabled by LEDGOs into the similarities and differences among the lytic enzyme domain repertoires commonly used by these bacteria, the architectural patterns connecting these building blocks in complete enzymes, and the amino acid sequences of the domains both overall and with respect to specific architectural contexts. LEDGOs also enabled discovery of a variety of

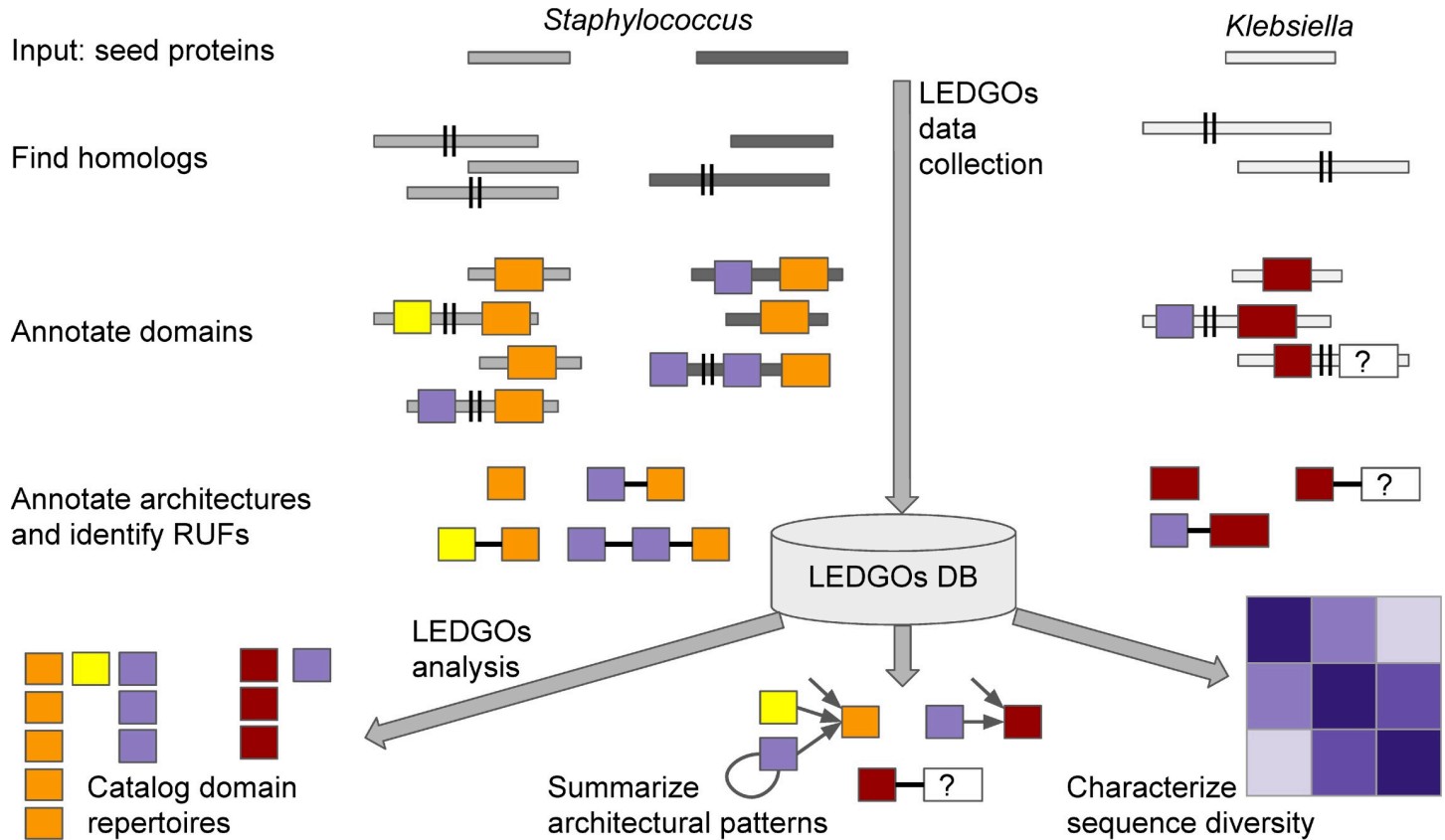

**Fig 1. LEDGOs workflow.** For each organism of interest, the user provides as input a set of "seed" proteins, here based on GO terms indicative of peptidoglycan recognition and catalysis. The LEDGOs data collection pipeline then gathers organism-specific homologs of the seed proteins by repeated PSI-BLAST searches. The LEDGOs pipeline further annotates the catalytic and cell-wall binding domains within the collected sequences according to Pfam families, and catalogs the domain architectures of the proteins. Note that the identified homologs can extend (past the marked "||") to include additional domains beyond those in the seeds. There can also be uncharacterized sequence regions (marked "?") between the annotated domains. The domain sequences, annotations, and architectures within full enzymes are stored in the LEDGOs database. The LEDGOs data analysis tools then query this database to characterize and compare/contrast the lysin domain building blocks and architectures employed by the different organisms.

unannotated "regions of unknown function", RUFs, situated in architectural contexts between characterized CAT and CWB domains in patterns suggestive of their potential functional utility.

**Table 1. LEDGOs database construction and composition.** A breakdown, by organism, of the counts of the initial sets of sequences with relevant annotation for peptidoglycan binding and catalysis function; representative seed sequences; identified sequence homologs; those homologs with catalytic domains; unique architectures; domain sequences; domain types; RUF sequences; and clustered RUFs.

| Organism | GO-annotated sequences | Seeds | Homologs | Homologs w/CAT | Architectures | Domains | Domain types | RUF sequences | RUFs |
|---|---|---|---|---|---|---|---|---|---|
| *Staphylococcus aureus* | 157 | 40 | 8811 | 1234 | 41 | 2239 | 23 | 2133 | 143 |
| *Enterococcus faecium* | 206 | 44 | 5108 | 706 | 41 | 1412 | 24 | 1074 | 145 |
| *Klebsiella pneumoniae* | 747 | 121 | 35009 | 1459 | 37 | 2059 | 28 | 1147 | 154 |
| *Acinetobacter baumannii* | 398 | 76 | 4608 | 869 | 29 | 1480 | 23 | 1011 | 116 |
| *Pseudomonas aeruginosa* | 272 | 66 | 20926 | 731 | 37 | 1149 | 27 | 774 | 116 |
| *Clostridioides difficile* | 447 | 118 | 3390 | 481 | 31 | 1001 | 17 | 977 | 144 |
| *Neisseria gonorrhoeae* | 21 | 8 | 166 | 49 | 8 | 78 | 8 | 56 | 12 |
| *Streptococcus pneumoniae* | 760 | 133 | 16151 | 534 | 38 | 1219 | 20 | 1658 | 214 |

## Common domains

We start by characterizing the frequency of different domain types, within each organism, among the lytic enzymes in the eight-pathogen LEDGOs database (Fig 2). To reduce overrepresentation of redundant sequences, we cluster the proteins to 95% identity and use only one representative sequence from each cluster. Then, among these sequences, we tally the occurrences of each different domain type. There are two complementary ways to evaluate frequency of a particular domain type: what fraction of the representative proteins contain that domain type at least once, and what fraction of the entire set of domain instances in the representative proteins are of that type. The second view assesses how much of a proteome's "mass" is of a given domain type, and can differ from the first view when, as is the case with some of the CWBs, a domain is commonly repeated two or more times in each protein containing it.

As discussed in the methods, we consider both specific domain families represented by a single sequence profile, along with superfamilies comprised of sets of domain families. For simplicity, we call both annotations "domain types" and we indicate the latter by appending an apostrophe to the end of the name in case the domain family and superfamily have the same name (cf. the domain family NLPC_P60 and superfamily NLPC_P60'). S1 Table summarizes the domain types that are represented in the following analyses, and generally characterizes their functions; we refer to the associated Pfam or CDD entries for further details.

The gram-negative bacteria rely on a relatively limited repertoire of CAT domains, dominated by N-acetylmuramidases and peptidases, in particular, members of the Lyz-like' superfamily (i.e., the Lysozyme_like superfamily) and the Peptidase_M23 domain, though notably *Neisseria* has fewer N-acetylmuramidases (note that the SLT domain is a member of the Lyz-like' superfamily). These two domain types are less frequently represented among lytic enzymes in gram-positive bacteria, though they do make up a noticeable fraction of the domains in *Enterococcus* and *Staphylococcus*, with *Clostridioides* and *Enterococcus* also leveraging to a lesser extent the specific Lysozyme_like domain itself.

Among the gram-positives, CHAPs dominate *Staphylococcus* and *Streptococcus* and are also quite common in *Enterococcus*, and these same three bacteria also commonly use Glucosaminidases. Notably, these two families of CATs are absent or nearly absent among our gram-negative bacteria. *Clostridioides* is a relative outlier among the gram positives, as its CATs are differentially dominated by Amidase_3 and NLPC_P60 members, though *Enterococcus* also makes some use of NLPC_P60.

A variety of MurNAc-LAAs are represented among both gram-positive and gram-negative CATs. Their role is largest in *Clostridioides* and smallest in *Acinetobacter*. Most bacteria use two or three of the four most common MurNAc-LAA families and superfamilies in the LEDGOs database, with *Enterococcus* and *Streptococcus* being the only species that employ peptioglycan recognition proteins (PGRP').

Turning to CWB domains, we see a preponderance of LysM among gram-negative bacteria, with *Acinetobacter* having the highest representation closely followed by *Neisseria*, while *Klebsiella* is seen with the lowest percentage. Here we see a difference between the frequency among proteins (bar charts) vs. among domains (waffle plots), with LysM found in a much larger proportion of lytic enzymes in *Neisseria* compared to *Acinetobacter*. This is driven by the number of repeated LysMs per enzyme, a topic we return to below. LysM is also the most common CWB in *Staphylococcus* and is utilized to a lesser extent by both *Enterococcus* and *Streptococcus*. The gram negatives complement LysM with a few other CWBs, notably AMIN in *Neisseria* and a bit in *Pseudomonas* and *Klebsiella*, and PG_binding_1 in *Pseudomonas*. The gram positives complement LysM with CWB types that are different from the negatives, and

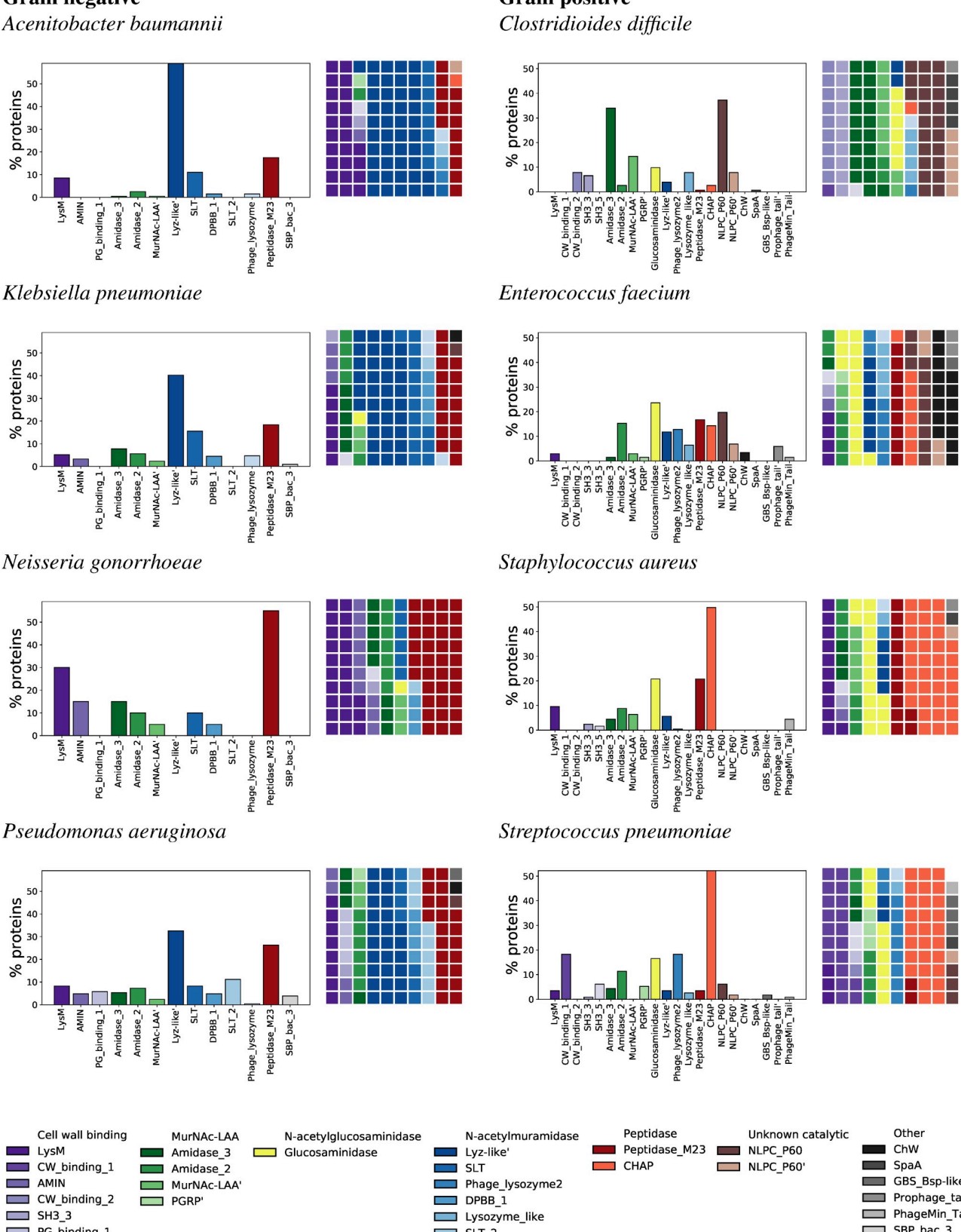

**Fig 2. Domain usage frequency by organism.** Bar charts indicate percentages of representative proteins containing each of the domain types. Waffle plots indicate percentages of each domain type among the set of domains comprising the representative proteins, separately counting duplicates of a domain type within a protein. With the entire set of domain sequences totaling 100%, each block in a waffle indicates that 1% of the domains are of a given domain type. Only domain types that appear in at least 2% of some organism's representative proteins are shown. Bars and waffle cells are ordered and colored by domain type as summarized in the legend.

in general from each other, most notably SH3_3 and CW_binding_2 in *Clostridioides*, and CW_Binding_1 and SH3_5 in *Streptococcus*.

*Enterococcus* is unique among our bacterial panel by its frequent use of Clostridial hydrophobic W (ChW) domains. While these domains show up in a relatively small proportion of enzymes (bar chart), it makes up a large percentage of the total domains due to repetition within each protein (waffle plot). While the function of the ChW family remains unknown, *Enterococcus* uses fewer of the common CWBs (a total of around 7% of domains) relative to the remaining bacteria (minimum 13% in gram-positive *Staphylococcus* and 11% in gram-negative *Klebsiella*, up to 26% in *Streptococcus* and 35% in *Neisseria*). This observation suggests that ChWs may encode peptidoglycan binding function.

## Architectural patterns

With the building blocks collected, we now leverage LEDGOs to analyze common architectural patterns in our set of bacteria—how these domains are connected together (or stand on their own) in lytic enzymes (Figs 3 and 4). The architectures characterized here appear in at least 2% of the proteins (as with domains, those representatives from the 95%-identity clusters) in at least one of the bacteria. To ensure that the proteins all have lytic activity, we limit analysis here to only those proteins containing at least one domain annotated as one of our CATs.

Almost all CATs are observed to stand on their own as lytic enzymes; i.e., a CWB is not required, though some CATs appear both with and without CWBs. One prominent CWB-dependent CAT among the gram negatives is SLT in *Neisseria*, which is always followed by LysM in these sufficiently frequent architectures (LysM optionally follows SLT in the other gram negatives). Another gram-negative CWB-dependent CAT is Amidase_3 in *Pseudomonas* and *Neisseria*, which is necessarily preceded by AMIN here (AMIN is optional in *Klebsiella* and Amidase_3 isn't represented in *Acinetobacter*). In the gram-positive *Clostridioides*, Amidase_3 is optionally preceded by one or more CW_Binding_2s.

The relative order of CATs and CWBs can vary. For example, CHAP in *Staphylococcus* can be preceded by LysM, whereas Amidase_2 in *Streptococcus* can be followed by CW_binding_1. In *Acinetobacter* alone, LysM is after SLT but before Peptidase_M23. This LysM points to one example of a 3-domain architectural pattern that we have not observed, despite having seen the constituent 2-domain architectures: SLT -> LysM -> Peptidase_M23. While not sufficiently frequent to appear in the figure, we have, however, observed a different "sandwich" of a CAT between two CWBs, AMIN -> Amidase 3 -> LysM in *Pseudomonas*, and likewise an infrequent CAT -> CWB -> CAT sandwich, Glucosaminidase -> LysM -> NLPC_P60 in *Enterococcus*.

Sometimes CATs pair up. For example, *Clostridioides* and *Enterococcus* both have a pair of CATs, Lysozyme_like (domain or superfamily) -> NLPC_P60, and *Clostridioides* also has CAT pairs Glucosaminidase -> NLPC_P60 while Staphylococcus has Glucosaminidase -> CHAP and its reverse, CHAP -> Glucosaminidase. CWBs most frequently pair with themselves in repeats, including LysM in *Acinetobacter*, *Neisseria*, and *Staphylococcus*; CW_Binding_1 in Streptococcus; and CW_Binding_2 in *Clostridioides*. There are also some low-frequency pairs not appearing in the figure, e.g., CHAP -> SH3_5 -> CW_Binding_1 in *Streptococcus*.

## Gram negative

*Acenitobacter baumannii*

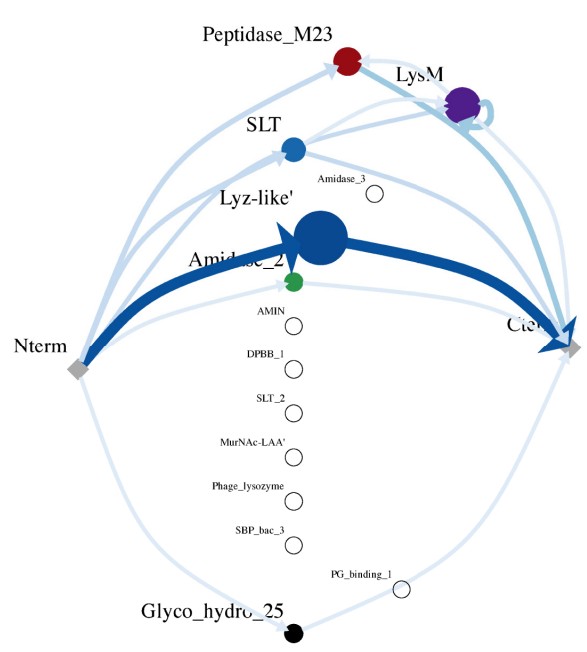

*Klebsiella pneumoniae*

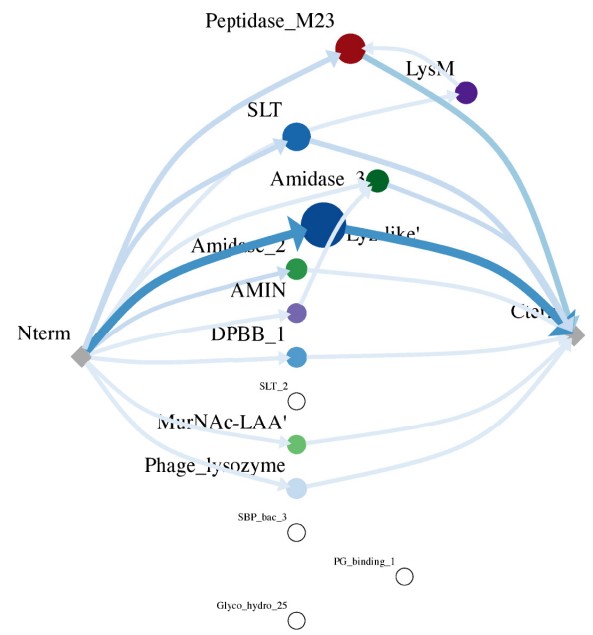

*Neisseria gonorrhoeae*

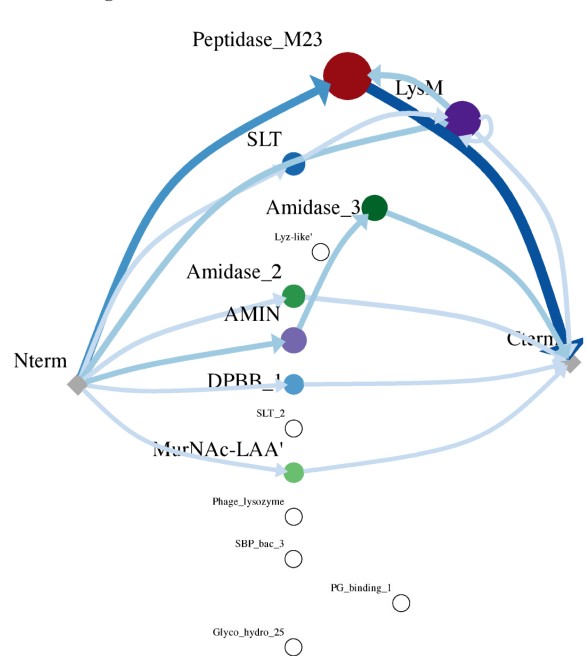

*Pseudomonas aeruginosa*

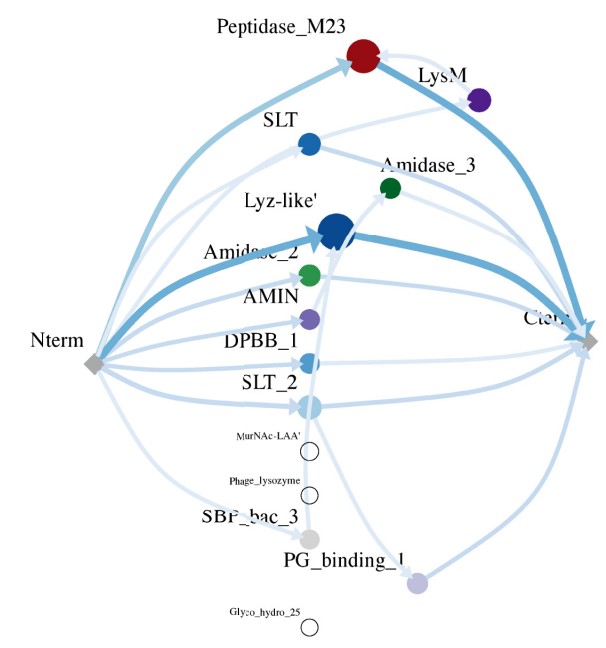

*% domain-domain pairs*

0   10   20   30   40   50

**Fig 3. Common gram negative lytic enzyme architectures.** In each graph, the nodes indicate domains (along with the N terminus and C terminus), with size reflecting frequency within the organism's clustered proteins and empty circles for domains with no representation in that species. The edges represent connections within a single protein, with edge shading and thickness representing relative frequency. Note that there are some self-edges (e.g., LysM loops back to itself), indicating a repeated domain. A path from Nterm through one or more domains to Cterm thus represents a protein, though not all such proteins have been observed in LEDGOs (see text).

## Domain sequence diversity

Having identified commonalities and differences in the types of and connections between domain building blocks comprising lytic enzymes in different bacteria, we now turn to characterizing the diversity within each domain type. That is, to what extent is the version of the domain in one organism similar to that in another, and in fact are there subgroups, perhaps with somewhat different specificities, within an organism? We summarize these patterns in heatmaps in Fig 5 for representative domains, and in S1 Fig for all those domains in Figs 3 and 4. While previous analyses used representative sequences, clustered to 95% identity, here we show all non-identical sequences in Fig 5, to give a sense of frequency of repetition along with levels of diversity.

Among the CATs, we see examples of domains that occur in a variety of these bacteria but whose sequences are quite diverse, suggesting that their recognition specificities may differ across the bacteria (SLT for gram negative, NLPC_P60 for gram positive, and Peptidase_M23 for both). Within these, there are often "sub-families" with higher conservation to each other within each organism. We also see some examples of domains that display relatively high identity across two or more species (Amidase_3 for gram negative; Lysozyme_like and NLPC_P60 for gram positive). The upper-right / lower-left dark blue block for Lysozyme_like indicates higher cross-species similarity than within-species similarity for these *Clostridioides*, *Enterococcus*, and *Streptococcus* sequences. Likewise, the upper-left block in Amidase 3, spanning all four gram negatives (though mostly *Klebsiella* and *Pseudomonas*) indicates relatively higher cross-species similarity. In contrast, there is little similarity among members of this family for the gram positives, and even within *Clostridioides* there are several distinct groups.

We hypothesize that some of the sub-families of similar domain sequences may be explained by the specific architectural context in which the domain appears. Fig 6 shows heatmaps for three of the domains, adding color coding for the rows and columns to indicate the architecture. For simplicity, we limit the analysis to just the common architectures as in Figs 3 and 4 (i.e., at least 2% of the representative proteins). To some extent, the blocks within Amidase_3 are associated with different architectures, e.g., distinguishing in the gram negatives whether it is on its own or with AMIN (with a couple blocks for each such architecture) and likewise distinguishing in the gram positives between Amidase_3 alone or with different numbers of CW_binding_2s. While overall CHAP is quite diverse, we see that the clusters are largely associated with different architectures, e.g., a block of CHAP -> Glucosaminidase in *Staphylococcus* and a block of Phage_lysozyme_2 -> CHAP in *Streptococcus*. Notably, there is little conservation between the CHAP domains preceded by Glucosaminidase to those succeeded by it in *Staphylococcus*. A striking example of architecture-related clustering is NLPC_P60, where we see that the sets of similar NLPC_P60s spanning *Clostridioides*, *Enterococcus*, and *Streptococcus* variants are in fact mostly preceded by a Lysozyme_like domain. This also explains some of the Lysozyme_like clustering observed in Fig 5 —those blocks are the ones associated with these NLPC_P60s.

Among CWBs, LysM is ubiquitous across both gram negative and positive, as discussed in the previous section (Fig 5). However, we see here that it is also quite species-specific, and even appears to have multiple specificities within each species. AMIN is common in both *Klebsiella*

## Gram positive

*Clostridioides difficile*

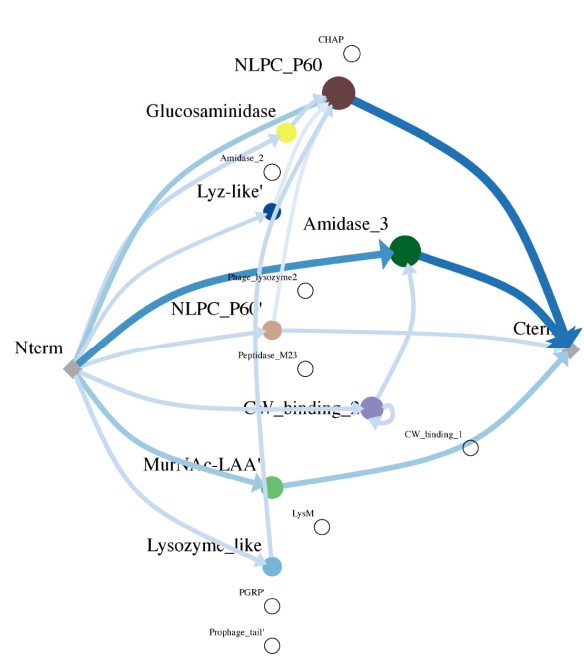

*Enterococcus faecium*

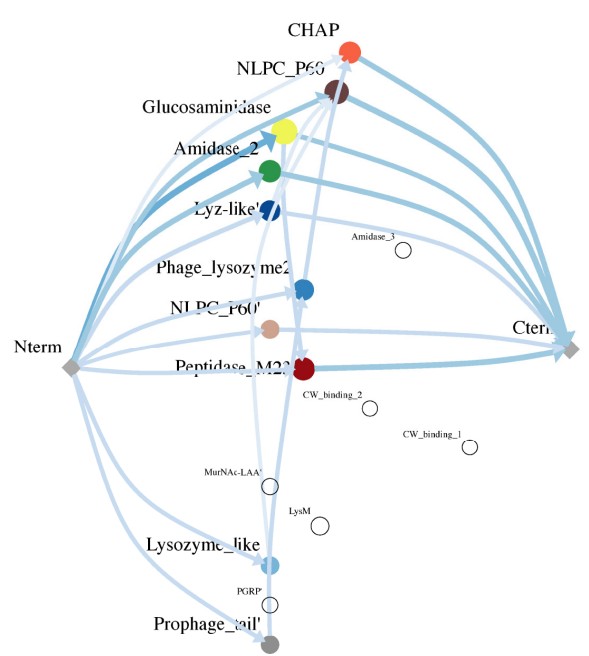

*Staphylococcus aureus*

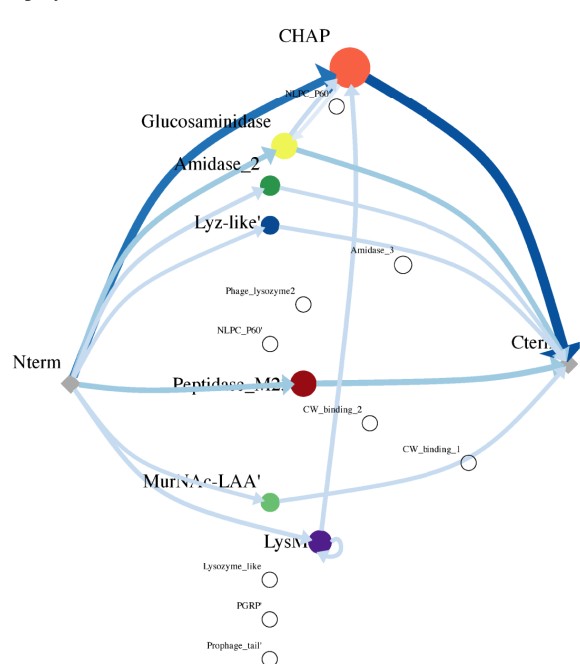

*Streptococcus pneumoniae*

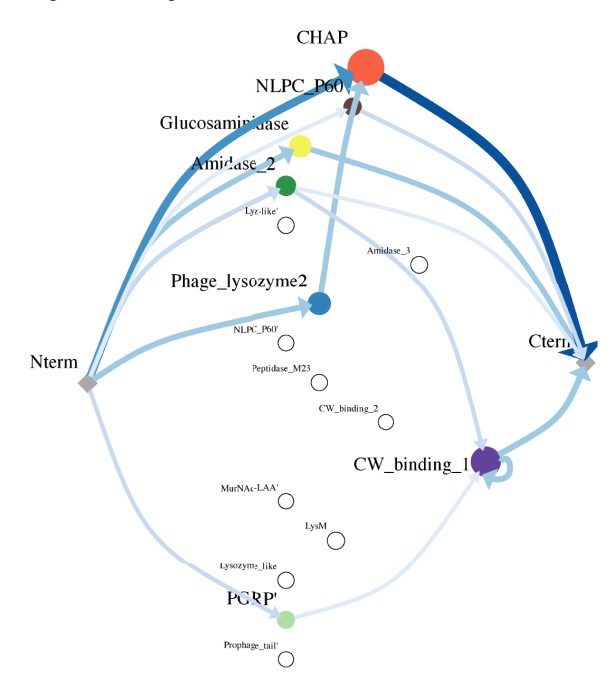

*% domain-domain pairs*

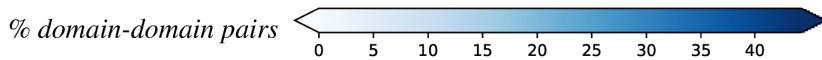

**Fig 4. Common gram positive lytic enzyme architectures.** In each graph, the nodes indicate domains (along with the N terminus and C terminus), with size reflecting frequency within the organism's clustered proteins and empty circles for domains with no representation in that species. The edges represent connections within a single protein, with edge shading and thickness representing relative frequency. Note that there are some self-edges (e.g., LysM loops back to itself), indicating a repeated domain. A path from Nterm through one or more domains to Cterm thus represents a protein, though not all such proteins have been observed in LEDGOs (see text).

and *Pseudomonas*, with some representation in *Neisseria*. We see fairly high identity between the second group of Klebsiella sequences with the *Neisseria* ones, and also somewhat higher between that group and the majority of *Pseudomonas* AMINs than with the other *Klebsiella* AMINs, although there is a *Pseudomonas* AMIN exhibiting striking identity to the first *Klebsiella* cluster. CW_Binding_2 only appears in *Clostridioides*, and it is clustered into several sub-families, which are the subject of the next section.

## Repeated domains

In order to characterize the extent of divergence of repeat domains within an organism, we used LEDGOs to generate heatmaps comparing the different copies. Fig 7 shows heatmaps for some representatives, in a format similar to that of Figs 5 and 6.

LysM occurs in groups of up to 10 repeats in *Acinetobacter*, though the vast majority of proteins have 8 repeats, in an architecture following SLT. In general, the sequences at a position are highly conserved, and there is less identity to other positions, though repeats number 5 and 6 are relatively more similar in *Acenitobacter*. We note that there are a few fairly rare architectures that show up as very thin rectangles of conservation.

In the gram positives, the majority of *Enterococcus* proteins using LysM have 6 repeats, and a sizable fraction of *Staphylococcus* have 2 or 3. In *Enterococcus*, all are highly conserved regardless of architecture. In *Staphylococcus*, a number of the position-2 LysMs are much more similar to the position-3 LysMs than to the other position-2 LysMs, and the position-1 LysMs are also split into two groups. This is driven by the number of repeats in the protein—the first cluster of position-1 and the first cluster of position-2 LysMs (along with the position-3 LysMs) are in architectures with three LysMs and then a CHAP. The position-3 LysMs show high similarity to their neighboring position-2 LysMs. The second clusters are in architectures with two LysMs and then a CHAP. As with the gram negatives, the very thin rectangles indicate rare architectures that have high conservation to the more common ones. Interestingly, here one such rare architecture has two LysMs followed by a "domain of unknown function", DUF 2286 superfamily (whose Pfam model is entirely based on archaeal sequences), and then a CHAP, suggesting that the DUF perhaps complements the LysMs' peptidoglycan-binding function.

CW_binding_2 in *Clostridioides* has up to three repeats, though there is only one instance with two repeats (thin rectangles in the heatmap). The conservation patterns for each repeat are the same, driven by the protein architecture they belong to—a small cluster with three CW_binding_2s and a Glucosaminidase and a larger cluster (with two quite similar subclusters) with three CW_binding_2s and an Amidase_3.

## Regions of unknown function (RUFs)

Finally, during the process of annotating domains of the proteins it finds, LEDGOs also notes the regions connecting them. Many of these regions are fairly short and are likely just linkers, but here we examine regions of at least 50 residues that are more likely to have some structure and function beyond connecting two domains. We consider only regions that were not annotated either by Pfam [47], which is what LEDGOs uses, or by NCBI's CDD [48], which

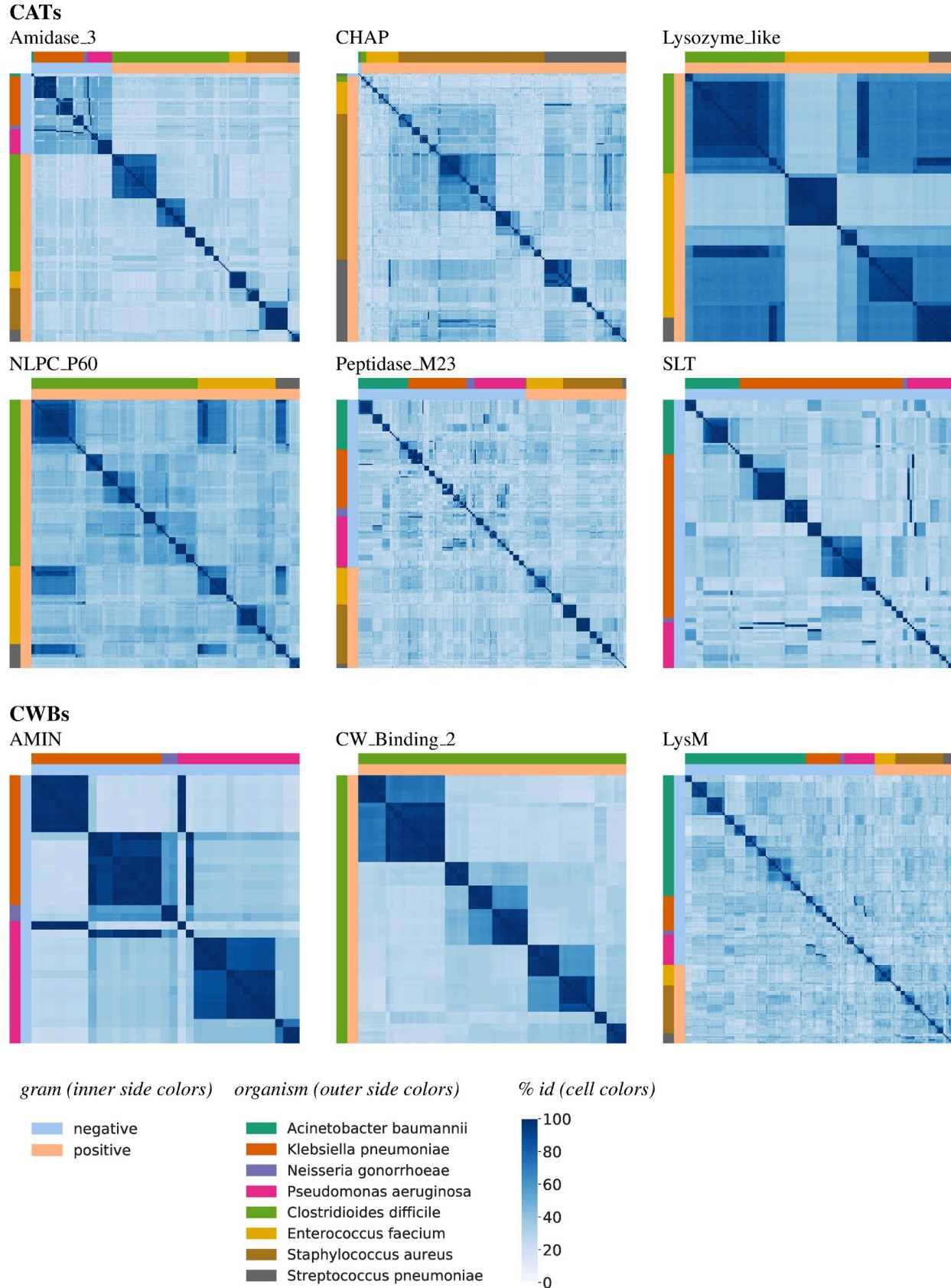

**Fig 5. Domain sequence diversity.** In each heatmap, each row and column represent a single non-redundant domain sequence in the LEDGOs database, and the cell for a pair of sequences is colored to indicate sequence identity (darker blue, higher). Cells are grouped by organism and clustered within an organism based on sequence identity patterns, so that similar sequences within an organism appear together as "blocks" on the diagonal, and blocks of similar sequences across organisms as off-diagonal blocks.

includes a number of other domain databases. All such unannotated regions of at least 50 residues, within sequences containing either a CAT or a CWB, are clustered at 40% global identity with a minimum length for each cluster member of 85% of the largest cluster member. Each cluster is identified as a region of unknown function (RUF); while we do not know if each region actually has one or more domains, we can examine the architectures in which it occurs to hypothesize possibilities. Fig 8 summarizes all RUFs with at least 50 occurrences and at least 75 residues (note that some occur in architectures with other RUFs that may or may not be in this top list); S4 Table provides representative accession numbers.

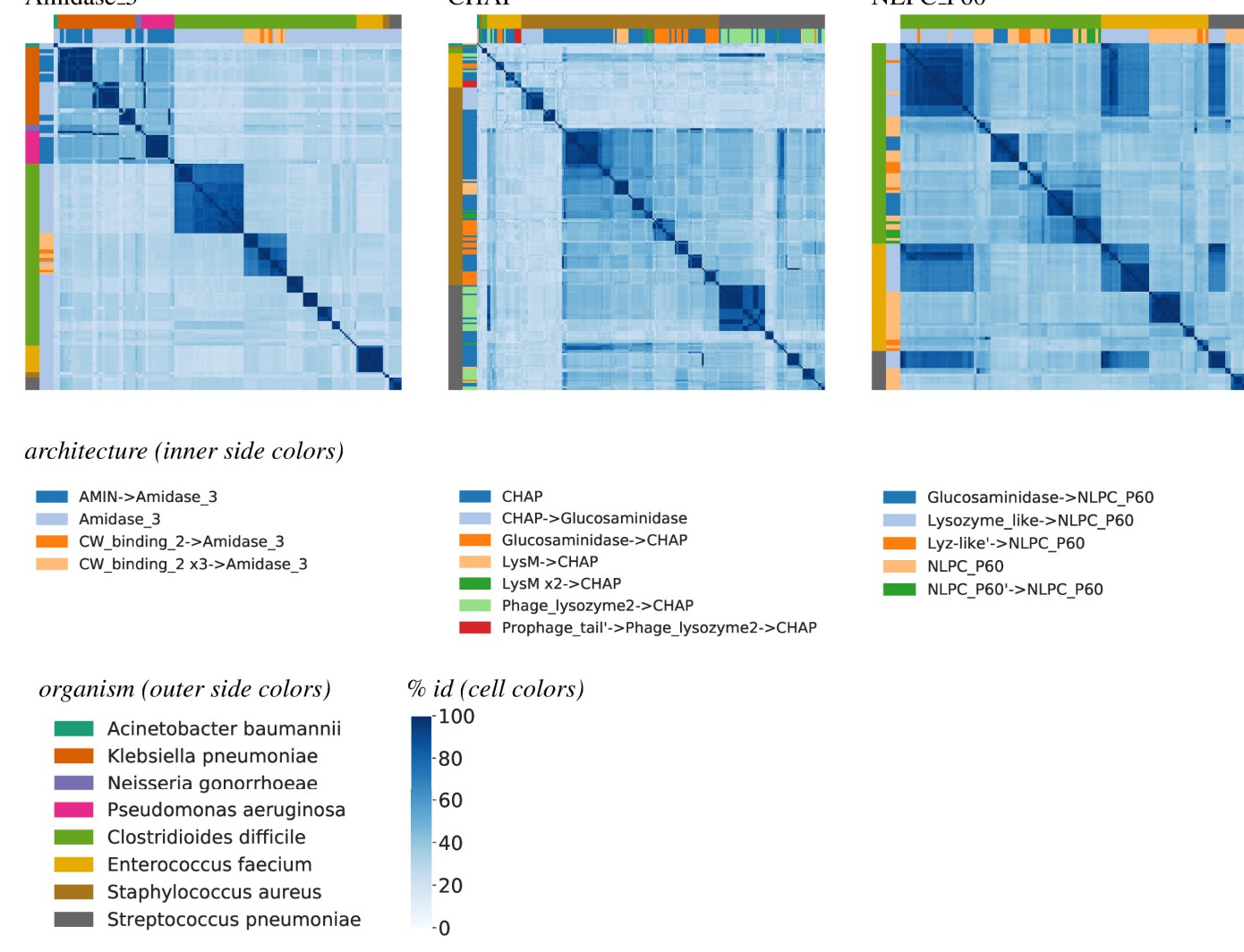

*architecture (inner side colors)*

- AMIN->Amidase_3
- Amidase_3
- CW_binding_2->Amidase_3
- CW_binding_2 x3->Amidase_3

- CHAP
- CHAP->Glucosaminidase
- Glucosaminidase->CHAP
- LysM->CHAP
- LysM x2->CHAP
- Phage_lysozyme2->CHAP
- Prophage_tail'->Phage_lysozyme2->CHAP

- Glucosaminidase->NLPC_P60
- Lysozyme_like->NLPC_P60
- Lyz-like'->NLPC_P60
- NLPC_P60
- NLPC_P60'->NLPC_P60

*organism (outer side colors)*

- Acinetobacter baumannii
- Klebsiella pneumoniae
- Neisseria gonorrhoeae
- Pseudomonas aeruginosa
- Clostridioides difficile
- Enterococcus faecium
- Staphylococcus aureus
- Streptococcus pneumoniae

*% id (cell colors)*

- 100
- 80
- 60
- 40
- 20
- 0

**Fig 6. Domain sequence diversity by architecture.** Heatmaps as in Fig 5, except limited to non-redundant domain sequences appearing in architectures with a frequency of at least 2% are shown, and with row/column colors indicating the organism and architecture rather than the organism and gram status.

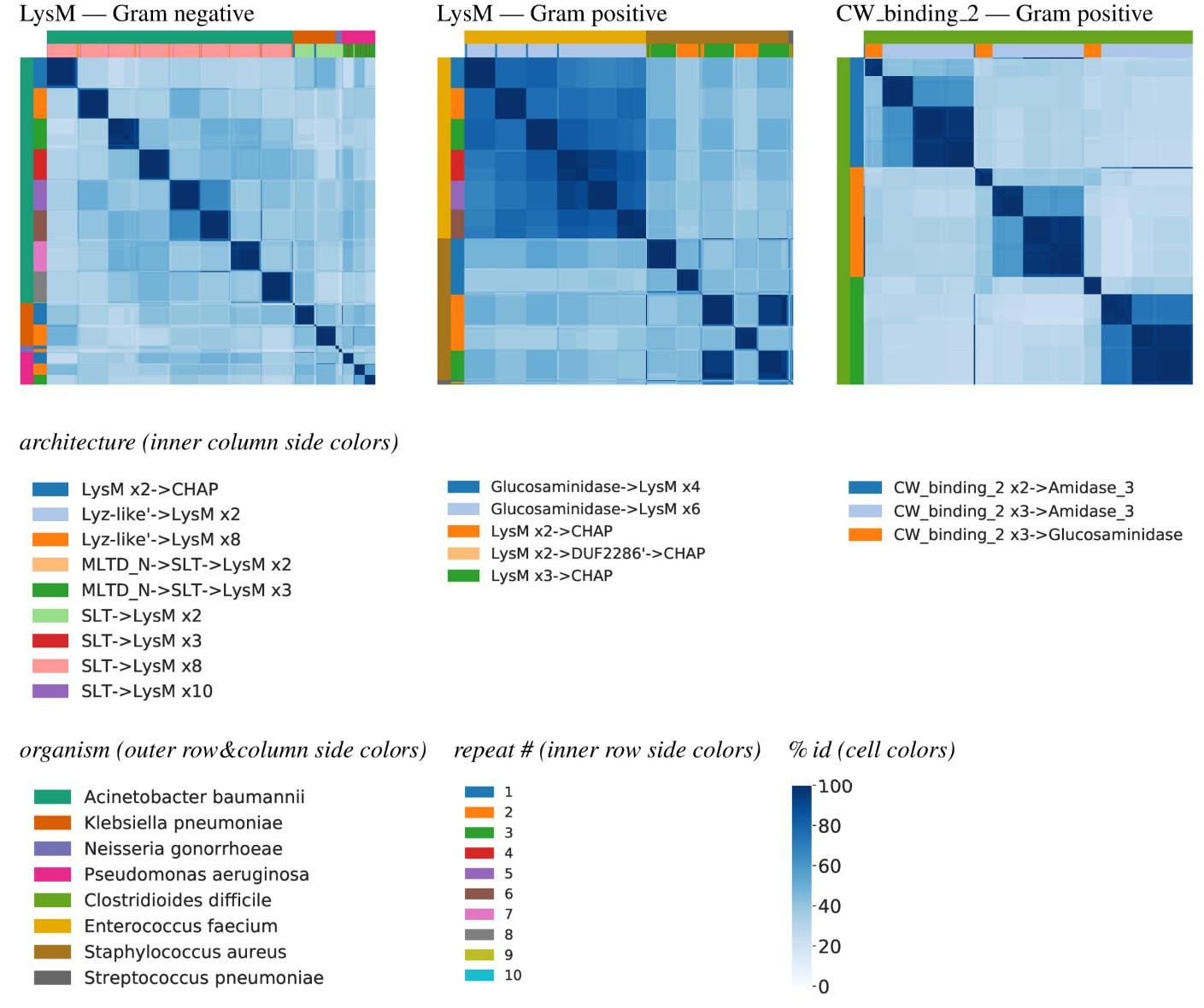

**Fig 7. Domain sequence diversity by repeat position.** Heatmaps as in Fig 5, except with the colors above the columns indicating the organism and architecture, and the colors beside the rows indicating the organism and repeat number. Thus cells are grouped by organism and repeat number, and clustered within those based on sequence identity patterns.

Based on their associations with either CWBs or CATs, we may hypothesize the functions of some RUFs. For example, RUF2, RUF6, and RUF8 are associated with amidases and thus perhaps serve as CWBs, while RUF1, RUF4, RUF5, RUF14, RUF16, RUF18, and RUF20 are associated with domains such as LysM and CW_binding_1 and are thus more likely to be CATs. A few of these top RUFs appear in more complicated architectures, with multiple annotated domains or with other RUFs. For example, RUFs 3, 7, and 11 are associated with an Amidase_2 and a Glucosaminidase; RUFs 11 and 12 are likewise associated with an SLT and a repeated LysM. All of these RUFs are of sufficient length to themselves contain one or more functional domains, though the functional utility of these domains is unclear.

| RUF | Seqs | Len | % Id | Organism | Architecture |
|---|---|---|---|---|---|
| 1 | 110 | 353 | 84.7 | Streptococcus | CW_binding_1, RUF1, Nterm, Cterm, YSIRK_signal RICH |
| 2 | 89 | 166 | 95.8 | Enterococcus | RUF2, Nterm, Cterm, Amidase_2 |
| 3 | 77 | 90 | 100.0 | Staphylococcus | Glucosaminidase, RUF3, SH3_8, Nterm, Cterm, RUF10, RUF7, Amidase_2 |
| 4 | 75 | 93 | 87.4 | Staphylococcus | RUF4, Nterm, LysM', LysM, Cterm |
| 5 | 64 | 128 | 65.9 | Enterococcus | RUF5, Nterm, LysM, Cterm |
| 6 | 70 | 99 | 73.2 | Clostridioides | RUF6, Nterm, Cterm, Amidase_3, MurNAc-LAA |
| 7 | 70 | 254 | 98.4 | Staphylococcus | See RUF #3 |
| 8 | 69 | 368 | 49.0 | Staphylococcus | Glucosaminidase, RUF8, Nterm, Cterm, CHA |
| 9 | 68 | 339 | 98.5 | Klebsiella | SPOR, Cterm, Nterm, RUF9 |
| 10 | 68 | 557 | 98.9 | Staphylococcus | See RUF #3 |
| 11 | 66 | 137 | 100.0 | Acinetobacter | LysM, RUF12, Cterm, Nterm, SLT, RUF11 |

| RUF | Seqs | Len | % Id | Organism | Architecture |
|---|---|---|---|---|---|
| 12 | 65 | 335 | 99.7 | Acinetobacter | See RUF #11 |
| 13 | 64 | 170 | 92.9 | Clostridioides | Amidase_3, Nterm, Cterm, CW_binding_2, RUF13 |
| 14 | 59 | 407 | 70.3 | Streptococcus | CW_binding_1, Nterm, Cterm, RUF14 |
| 15 | 56 | 120 | 97.5 | Staphylococcus | Peptidase_M23, Trep_Strep', RUF15, PhageMin_Tail, Nterm, Cterm, RUF110, DUF4200, RUF123, RUF9 |
| 16 | 54 | 280 | 98.6 | Acinetobacter | RUF16, Nterm, LysM, Cterm |
| 17 | 53 | 290 | 96.4 | Streptococcus | RUF17, Nterm, LysM, Cterm |
| 18 | 52 | 494 | 62.7 | Streptococcus | CW_binding_1, Nterm, Cterm, RUF18 |
| 19 | 51 | 148 | 98.6 | Streptococcus | RUF19, CW_binding_1, Nterm, Cterm, Lactamase_B, metallo-hydrolase-like_MBL-fold, RUF142 |
| 20 | 51 | 407 | 96.7 | Enterococcus | NLPC_P60, Cterm, Nterm, RUF20 |
| 21 | 50 | 132 | 96.0 | Staphylococcus | LysM, RUF21, Cterm, Nterm, B, YSIRK_signal |

**Fig 8. Regions of Unknown Function, RUFs.** Entries give each RUF's number of sequences, median sequence length, median pairwise sequence identity, organism in which it appears, and architecture graph (as in Figs 3 and 4) constructed from all architectures occurring at least 5 times.

Since, as we have seen, it is common for CWBs to repeat, we checked for that possibility in some of the putative CWB RUFs. RUF8 is of particular interest as it is situated between a CHAP and a Glucosaminidase, and it has roughly 372 residues with fairly high sequence diversity over the 67 proteins in which it appears. A RADAR (Rapid Automatic Detection and Alignment of Repeats) [49] analysis identifies two potential sets of repeats (S2A Fig), with the larger one from residues 202 to 242 and 335 to 369—small, but about the same size as LysM so perhaps sufficient for CWB activity. Searching LEDGOs with the representatives of the two versions identifies 3 additional sequences, all truncated in some form compared to the initial architecture. Overall, we see substantial sequence diversity between these two subsequences (S2B and S2C Fig), with an average sequence identity between the two repeats of 22%, compared to 68% within the first one and 96% within the second one. Thus, while this analysis represents a promising way to attempt to decompose RUFs into sub-RUFs, this result suggests that, if these are indeed duplicated, homologous CWBs, they have substantially diverged in their specificities.

We further analyzed these RUFs to determine if they appear in other genera. All RUFs summarized in Fig 8 were BLASTed [50] (0.005 E-value cutoff) against NCBI's Non-redundant (nr) BLAST database [51]. The BLAST results with at least 40% identity and a minimum length of 85% of the query sequence were counted by genera and those with at least 50 hits are summarized in S4 Table. Of the 21 RUFs, 3 appeared at least 50 times in multiple genera. Of particular note is RUF9 which is seen in 7 genera, most significantly in Salmonella (1019), Escherichia (656), and Klebsiella (650). The representative sequence of RUF9 is associated with the cell division protein DamX which anchors to peptidoglycan via the Sporulation related (SPOR) domain [52]. DamX has been shown to play a role in bile resistance in Salmonella enterica [53] and cell division in *Escherichia coli* by stimulating transglycosylase/transpeptidase activity of penicillin-binding protein 1B (PBP1B) [54,55]. Due to the peptidoglycan binding function of SPOR, we can hypothesize that RUF9 plays a role in PBP1B stimulation and may play a similar role in other species.

## Discussion

Bacteria make extensive use of autolysins to remodel, recycle, and even destroy their cell walls. Systematic, comprehensive investigations into autolysin repertoires, as we have initiated here, can provide insights into these fundamental biological processes, as well as how they can be controlled with targeted chemotherapeutics or leveraged in an aikido-like fashion, turning autolysins against their own sources as antibacterial agents. The LEDGOs pipeline we have developed here has enabled us to quickly move far beyond our initial goal of finding *S. aureus* autolysins and scale up to characterize diversity in autolysin repertoires across a diverse set of pathogenic bacteria. It promises to likewise enable facile application to a wide range of genera, species, and even strains. LEDGOs essentially serves as a discovery tool, enabling subsequent bioinformatic and experimental investigations to be focused on proteins, domains, and architectures that are most significant and relevant to the task at hand. Our analyses of autolysin building blocks in eight important pathogens has already revealed intriguing patterns of sequence diversity, domain usage, and relationships among the domains comprising their lysins.

The LEDGOs pipeline automatically performs a thorough search for, and annotation of, organism-specific autolysin domains based on GO terms, PSI-BLAST sequence searches, and

Pfam domain identification via NCBI's CDD. Within this pipeline, we have earmarked some variations for future development. First, since bacterial proteomes are fairly small and LEDGOs performs repeated PSI-BLAST searches from multiple initial templates within each organism, we are likely to discover the proteins most likely to have autolytic function (along with others that may not). However, it is possible that additional, even-more-remote homologs could be found by specialized search techniques [56,57] or by separate searches for individual domains with profiles combined across organisms. Second, in annotating domains, for consistency and simplicity we employed only Pfam searches (except when double-checking RUFs). CDD search can also account for additional domain annotation databases beyond Pfam, namely SMART [58], COG [59], PRK [60], and TIGRFAMs [61], though the combination of annotation services can lead to overlapping, inconsistent annotations which would need to either be presented as alternatives or rectified. Finally, while LEDGOs is freely available and provides an automated process for collecting and analyzing autolysins from a new organism, it does require familiarity with installing the associated tools and running scripts from the command-line. Thus, with the foundations now firmly in place, an important next step is to establish a web tool supporting end-user construction and analysis of organism-specific and cross-organism lysin domain building blocks and architectures.

The analyses here focus primarily on frequent domains and architectures, setting thresholds for minimum relative representation within proteomes. This characterization of common patterns sets the stage for further analysis of more rare "outliers", which may be artifacts of sequencing, or may in fact hold interesting biology or utility for engineering. For example, the fairly rare glycoside hydrolase family Glyco_hydro_25 contributes to some *Acinetobacter* architectures. Likewise, while it is relatively uncommon to have two different types of CAT in the same enzyme, we did observe some instances such as Lysozyme_like -> NLPC_P60 (*Clostridioides* and *Enterococcus*), Glucosaminidase -> NLPC_P60 (Clostridioides) and both Glucosaminidase -> CHAP and CHAP -> Glucosaminidase (Staphylococcus). With CWBs we saw AMIN -> Amidase 3 -> LysM (*Pseudomonas*) and an infrequent CHAP -> SH3_5 -> CW_Binding_1 (*Streptococcus*). Further analyses of these could reveal how bacteria are leveraging multi-functionality / multi-specificity. While we saw variable numbers of repeats of some CWBs (e.g., up to 10 LysMs), the distributions are uneven. Presumably some of this is due simply to how evolution unfolded, along with biases in the sequencing record, but further investigation could evaluate the relative utility of different numbers of repeats, and whether particular sequence variations among the repeats are being used to broaden or narrow specificity, provide avidity, or are just artifacts. One important aspect in all of this is that "frequency" in the current incarnation of LEDGOs is based on the number of occurrences in the sequence record. Complementing this information with gene expression or protein quantification data, where available, could be very informative in assessing the relative impact of the raw numbers of coded sequences on different biological processes at different points in bacterial life cycles.

Our analysis of RUFs evaluated conservation of these uncharacterized sequence regions across the organisms in the LEDGOs database and looked for repeat patterns within them. We expect that, with continuing updates of sequence databases, some of these RUFs may receive annotations; in the mean-time, the RUF analysis suggests sequences worth further exploring. To go further in attempting to infer functional roles for these regions, we could more generally search for remote homologs in the broader genomic database [62], employ a specially-trained predictor of lytic function [63], or see if predicted structures [64] provide any insights. We could also look at sequence patterns that might be indicative of disorder [65], membrane association [66], or other general properties that might suggest what these regions molecular function, if any. We limited RUFs to at least 75 amino acids in length, so presumably they are not simply inter-domain linkers. However, a similar analysis of sequence patterns of actual inter-

domain linkers may also be interesting, and reveal "rules" for how domains are connected together in a not-too-rigid but not-too-flexible manner [67–69].

In expanding beyond the eight pathogens represented in the initial LEDGOs database, it would be beneficial to go both broader and deeper. For breadth, a natural extension would be to bring in other sources of lysins, including bacteriophage, since some bacterial autolysins are derived from genomically integrated phage endolysins and exolysins [70,71]. Incorporating the sequence, domain, and architecture patterns of other lysins (e.g., phage endolysins, animal lysozymes, bacterial bateriocins) may help in searching for autolysins, shed light on divergence of function, provide another point of reference for comparing and contrasting autolysin patterns, and potentially serve as blueprints for deriving new molecules. For depth, while we focus here on only one species from each bacterial genus, comparison to their siblings or cousins might be informative for the same reasons. For example, autolysin building blocks and architectures from *Staphylococcus simulans* could provide a useful reference point for evaluating significance of those in *Staphylococcus aureus*.

LEDGOs can be used to identify potential drug targets, i.e., enzymes whose inhibition would be detrimental to bacterial physiology, as has been shown with knock-outs of LytN and Atl [22,26]. If the goal is to develop a broad-spectrum drug, one should consider a highly popular node in many architectures (Figs 3 and 4), with reasonable sequence conservation across the relevant species (Fig 5). For example, some NLPC_P60s are relatively conserved across *Clostridioides*, *Enterococcus*, and *Streptococcus*, and some AMINs display high sequence identity between *Klebsiella* and *Neisseria*. If, on the other hand, a narrow spectrum agent is required, then a species-specific architecture and/or sequence pattern would be desired. There are plenty of such CHAP and LysM domains. In both cases, as mentioned above, it would be helpful to incorporate gene expression and protein abundance data to further assess the importance of the identified domains.

The LEDGOs-based analysis of putative autolysins, and their constituent domains and architectural patterns, could provide valuable guidance toward engineering autolysin-based drugs. These drugs co-opt the pathogens' own essential enzymes as potent, efficient antibiotics which, as part of natural and essential bacterial physiology, may be more refractory to resistance than traditional small-molecule drugs. In terms of relative utility of a domain, along with its spectrum of selectivity, the analysis is much like that for identifying drug targets. But in building a biotherapeutic, we can move beyond natural lysins by treating their domains as building blocks that can be recombined in new and potentially more potent ways [28,29,39]. For example, a version of a CAT that doesn't have a CWB, or rarely does, could be augmented with a CWB that works well with another CAT for that organism, or with a type of CWB that works well with that type of CAT in a different organism. Additional CWBs could be identified by lifting the restriction imposed here to proteins that contain CATs. Bi- or multi-functional architectures could be assembled, e.g., combining into a new "sandwich" two CATs that work on opposite sides of the same CWB in different proteins, as we suggested re SLT -> LysM + LysM -> Peptidase_M23 (*Acenitobacter*). This same approach could lead to avidity or finer control of breadth by combining CWBs that have different peptidoglycan recognition specificities (e.g., an SH3_5 and a LysM). CATs could also be combined based on understanding of functionality; e.g., the both the lytic transglycosolase DPBB_1 and an amidase are used for cell separation [72], and could be integrated into a single antibiotic enzyme.

## Methods

LEDGOs is implemented as a set of python and shell scripts, available from the git repository at https://git.dartmouth.edu/cbklab/ledgos. The database is stored in SQLite, accessed via the

python package SQLAlchemy [73]. The general structure of the database is outlined in S3 Table; each step along the pipeline queries and/or adds to this set of tables as appropriate. A dump of the database created and used for the results section is also available in the git repository.

The LEDGOs data collection pipeline processes each organism's sequences separately, starting from an organism-specific set of sequences. This makes it "plug and play" to instantiate LEDGOs with a new organism; all subsequent processing steps are the same, as summarized in Fig 1. Here we walk through the concrete processing steps that produced the presented results, broken down into steps to load data into the database, followed by steps to use the database to analyze domain and architecture patterns.

## Database creation

**Collect organism-specific sequences.** All protein sequences for the organism are downloaded from UniProtKB via their REST API [74]. An organism-specific BLAST database [50] is constructed to use in searching those sequences.

**Identify seed sequences.** The collection of putative lytic enzymes and their constituent domains starts with proteins that are already annotated with GO [75,76] terms associated with peptidoglycan binding and catalysis, e.g., "peptidoglycan binding", "peptidoglycan endopeptidase activity", and "peptidoglycan muralytic activity". The current terms are provided in S2 Table. All identified sequences and their metadata are stored in the LEDGOs database. The sequences are clustered to 90% identity using CD-HIT [77], with the cluster representatives then serving as the seeds.

**Find homologs.** Each seed sequence is PSI-BLASTed [78] (3 iterations, 0.005 E-value cutoff) against the organism-specific database. The returned *full* sequences (i.e., not just the parts locally aligned against parts of a seed) and their metadata are stored in the database, with unique entries by accession number. The full sequences may contain additional associated domains not represented among the seed sequences. Thus to cast an even broader net, this expanded set of sequences is clustered to 90% identity using CD-HIT, and the PSI-BLAST process is repeated once more with this larger set of representative sequences.

**Annotate constituent domains.** Possible domains within each sequence are annotated via command-line RPS-BLAST (0.01 E-value cutoff) [79] against the Pfam profile database [47] as supplied with NCBI's Conserved Domain Database (CDD) [48]. The most significant non-overlapping domains within each sequence are identified by the CD-search post-processing utility rpsbproc (0.01 E-value cutoff). Rather than indicating that a portion of a sequence is an instance of a specific type of domain, an annotation could indicate only that it belongs to a domain *superfamily*, which is generally defined as a set of sequence-similar domains assumed or known to be functionally related. rpsbproc distinguishes these two levels of annotation based on how well a sequence matches a domain profile: a specific domain type is assigned when the sequence matches the domain profile to within a profile-specific threshold, while a more general superfamily type is assigned when the sequence matches a domain profile but does not meet the domain-specific threshold. For simplicity, we label each form of annotation as a "domain type", appending an apostrophe to the name of a superfamily annotation. Thus for example, NLPC_P60 and NLPC_P60' can both indicate that the sequence matches the NLPC_P60 domain profile, with the former meeting a threshold that more definitively indicates it is that type of domain and the latter suggesting it could be an instance of that domain or something closely related. The domain types are stored as individual entries, with associations linking a protein to its constituent domain types including the start and stop residues of the domain types within the protein.

Domain types themselves are classified into CAT, CWB, and Other using both Interpro's GO terms [80] for the domain types, along with manual curation of domain types inferred from CDD's functional descriptions. In addition, a domain type is inferred to be a CAT or CWB if another member in the same superfamily is annotated to be one. Likewise, a superfamily is inferred to be a CAT or CWB if one of its domains is known to be such. CAT domain types are further classified based on their catalytic target (MurNAc-LAA, N-acetylglucosaminidase, N-acetylmuramidase, Peptidase, and Unknown catalytic) according to [21]. S1 Table summarizes the resulting annotations for the presented results.

**Annotate architectures.** Once constituent domains have been annotated, each protein's architecture is classified as an ordered list of domain types; e.g., LysM->CHAP, Glucosaminidase->CHAP, and LysM->LysM->CHAP. The architectures are pieced together from their constituent domains in the database.

**Identify regions of unknown function, RUFs.** When the final residue position in one annotated domain is sufficiently far away from the initial residue position in the next annotated domain, we label the intermediate residues as a RUF. Short inter-domain spacing is presumably just a linker, but longer RUFs may be as-yet-undiscovered domains; the results presented here used a minimum RUF size of 50. (We call them "regions" as we don't yet know whether they are truly domains, or maybe even contain multiple domains.) To ensure that a RUF is truly unannotated, we expand the annotation search beyond Pfam to all annotation services provided in CDD.

To identify similar RUFs, the sequences are clustered with CD-HIT using a global identity threshold of 40% and a length difference cutoff of 85%. Each cluster is given a unique identifier (RUF1, RUF2, etc.) and an associated reference sequence region based on the accession number and residue positions of the CD-HIT cluster representative. Much like domains, RUFs are stored as individual entities, including their unique identifier and associated reference sequence region. Associations between proteins and their RUFs are stored, indicating the start and stop residues in the proteins.

## Data analysis

**Representative proteins.** In calculating frequencies of occurrence of domains and architectures, overrepresentation of redundant sequences can bias the calculations. In order to reduce this bias and compute statistics from more diverse representatives, the sequences are pre-clustered using CD-HIT [77] at a fixed threshold, and from each cluster only the representative sequence is used. Results were generated at 90%, 95%, and 99% identity, and the same general patterns emerged. The results presented here use 95% identity.

In order to ensure that the proteins being analyzed are lytic enzymes and not just peptidoglycan-associated proteins, sequences are pre-filtered to those containing an annotated CAT domain, for all analyses other than the assessment of RUFs, as these regions might provide such catalytic function.

**Domain frequencies.** Domain frequencies are assessed from two different viewpoints. The first viewpoint considers how many "copies" of a particular domain are necessary to construct all the representative proteins. Frequencies are computed as the number of instances of a domain divided by the total number of domains over all representative proteins; they sum to 1 and can thus be presented as a waffle chart (generated here using the python pywaffle package [81]). The second viewpoint considers how many of the representative proteins contain a particular domain at all. Here, frequencies are computed as the number of representative proteins with one or more copies of the domain, divided by the number of representative proteins. These frequencies are presented as bar charts.

**Architecture frequencies.** The frequency of an architecture is computed as the fraction of the representative proteins with exactly that architecture. The frequency of a domain A -> domain B connection is computed as the fraction of the representative proteins containing that pair of domains in that order. For clarity of visualization, architectures are filtered for sufficient frequency, and the domains and edges are presented as graphs drawn by the Graphviz software [82] via the python pygraphviz package [83], with a custom script setting graphical parameters and laying out domains on the page according to frequency of the position of the domain between N- and C-terminus.

**Sequence diversity.** Sequence diversity plots use all unique sequences rather than just the representatives, in order to give a sense of the overall commonalities and differences in the proteomes. The same general analysis is performed for variants of each domain, for variants by repeat position, and for RUFs. Sequences are grouped (by organism, repeat position, etc.) and clustered by percent identity using the fastcluster python package [84]. Heatmaps are generated using the python matplotlib [85] and seaborn [86] packages.

## Supporting information

**S1 Fig. Domain sequence diversity, for all domains in the presented LEDGOs database.** In each heatmap, each row and column represents a single non-redundant domain sequence in the LEDGOs database, and the cell for a pair of sequences is colored to indicate sequence identity (darker blue, higher). Cells are grouped by organism and clustered within an organism based on sequence identity patterns, so that similar sequences within an organism appear together as "blocks" on the diagonal, and blocks of similar sequences across organisms as off-diagonal blocks. Architectures of the associated sequences are indicated by row/column colors.
(PDF)

**S2 Fig. Analysis of repeats in RUF8.** (A) Output from RADAR highlighting two sequence regions with similar sequences. (B) Sequence identity between RUF-8 repeat regions 1 (blue side color) and 2 (orange side color). (C) Sequence logo for the RUF-8 repeat sequences.
(PDF)

**S1 Table. Domain types used in the presented LEDGOs database.**
(PDF)

**S2 Table. GO terms used in the presented LEDGOs database.**
(PDF)

**S3 Table. Simplified LEDGOs database schema.**
(PDF)

**S4 Table. BLAST search of nr for selected RUFs.**
(PDF)

## Author Contributions

**Conceptualization:** Spencer J. Mitchell, Deeptak Verma, Karl E. Griswold, Chris Bailey-Kellogg.

**Data curation:** Spencer J. Mitchell.

**Formal analysis:** Spencer J. Mitchell, Chris Bailey-Kellogg.

**Funding acquisition:** Karl E. Griswold, Chris Bailey-Kellogg.

**Investigation:** Spencer J. Mitchell, Karl E. Griswold, Chris Bailey-Kellogg.

**Methodology:** Spencer J. Mitchell, Deeptak Verma, Chris Bailey-Kellogg.

**Software:** Spencer J. Mitchell, Chris Bailey-Kellogg.

**Supervision:** Chris Bailey-Kellogg.

**Visualization:** Spencer J. Mitchell, Chris Bailey-Kellogg.

**Writing – original draft:** Spencer J. Mitchell, Chris Bailey-Kellogg.

**Writing – review & editing:** Deeptak Verma, Karl E. Griswold.

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
