## [Decision Letter · Decision Letter 0]

9 Feb 2021

Dear Dr. Bailey-Kellogg,

Thank you very much for submitting your manuscript "Building blocks and blueprints for bacterial autolysins" for consideration at PLOS Computational Biology. As with all papers reviewed by the journal, your manuscript was reviewed by members of the editorial board and by several independent reviewers. The reviewers appreciated the attention to an important topic. Based on the reviews, I am likely to accept this manuscript for publication, providing that you modify the manuscript according to the review recommendations.

Sincerely,

Nir Ben-Tal

Deputy Editor

PLOS Computational Biology

[LINK]

Reviewer's Responses to Questions

**Comments to the Authors:**

Reviewer #1: The manuscript by Mitchell et al. reports a new analysis tool for bacterial autolysins. The LEDGOs pipeline allows for a comparison of autolysins and their predicted domain architectures across bacterial species. This will be a highly useful tool for the field. The paper is well-written and informative and I only have minor comments.

1. Line 80 – spell out CHAP upon first use

2. Line 92 and elsewhere – the authors often semantically conflate using autolysins as antibiotics with using them as novel antibiotic targets. In line 92, it should be antibiotic targets.

3. Line 96 – for completeness’s sake, also cite the Gram-negative literature, e.g. PMID 23062283, 31286580, 26032134,

4. Line 100 should be “biotherapeutic target”

5. Line 119 etc: Please outline how these pathogens were prioritized for analysis vs. the more standard model organisms E. coli and B. subtilis (the ones they use are actually better, but for readability it should still be spelled out clearly why they abandoned the model organisms)

6. Line 198 spell out PGRP upon first use

7. Lines 207 and 209 – stay a bit more formal, i.e. spell out “Gram-negatives” and Gram-positives”

8. Line 213 spell out ChW domains upon first use.

9. Line 281 seems like a contradiction – some domains occur in a variety of bacteria but appear to be specific to each? (specific suggests they only occur in certain bacteria, not a variety). Please make clearer. Also, as far as I know, NlpC_P60 enzymes occur in both Gram-positives and Gram-negatives (E. coli’s MepS is a P60 family enzyme – but maybe the annotation has simply changed now?).

10. Fig. 7 would be even more useful to the field if they gave representative gene names/uniprot accession for these RUFs

Reviewer #2: Mitchell et al. introduce LEDGOs, a pipeline to find and characterize autolysins with the aim to find new chemotherapeutic targets and to develop antibacterial biotherapies. The authors apply LEDGOs to eight different pathogens and following they describe and compare the results between the different organisms in regard to their domain composition, domain architecture as well as sequence diversity. Furthermore, they analyse interdomain regions of unknown function and group the potential new domains as rather catalytic or cell wall binding domains.

Introduction: The authors may want to introduce LEDGOs in more depth already in the introduction by differentiating between the LEDGOs pipeline and LEDGOs database and by explaining how beneficial this pipeline could be for the users.

Fig. 1: It is useful to have an overview figure (Fig. 1), which explains how LEDGOs works but I may recommend re-structuring it. It is not clear which texts and images belong to the long arrow and particularly which part of the analysis is done by LEDGOs. Based on the current figure I would assume that I need all information (seed proteins, homologs, domains, architecture, RUFs) before running LEDGOs and not that this information will be stored in the LEDGOs database.

Fig. 6: Is the legend description in figure 6 correct that the row colors indicate the repeat numbers and the column colors denote the architecture? I assumed that the inner left side is called ‘inner column’ and represents the repeat number and the inner top row is called ‘inner row’ and represents the architecture.

Page 16, line 351-354: The authors may consider adding the DUF 2286 domain plus PFAM ID to the supplementary table S1. It might also be interesting to note that in PFAM, DUF 2286 is so far found only in archaea.

Fig. 7: Even though the analysis was conducted for different bacterial pathogens separately, it would be interesting to know in how many different bacterial genera the identified RUFs can be found and if the RUFs are found in other protein types besides autolysins.

**Have all data underlying the figures and results presented in the manuscript been provided?**

Reviewer #1: Yes

Reviewer #2: None

PLOS authors have the option to publish the peer review history of their article (what does this mean?). If published, this will include your full peer review and any attached files.

Reviewer #1: No

Reviewer #2: No
---

## [Editor Report · Decision Letter 1]

17 Mar 2021

Dear Dr. Bailey-Kellogg,

We are pleased to inform you that your manuscript 'Building blocks and blueprints for bacterial autolysins' has been provisionally accepted for publication in PLOS Computational Biology.

Best regards,

Sergei L. Kosakovsky Pond, PhD

Associate Editor

PLOS Computational Biology

Nir Ben-Tal

Deputy Editor

PLOS Computational Biology

---

## [Editor Report · Acceptance letter]

26 Mar 2021

PCOMPBIOL-D-20-01785R1 

Building blocks and blueprints for bacterial autolysins

Dear Dr Bailey-Kellogg,

I am pleased to inform you that your manuscript has been formally accepted for publication in PLOS Computational Biology. Your manuscript is now with our production department and you will be notified of the publication date in due course.

With kind regards,

Katalin Szabo
